# Compost Quality Recommendations for Remediating Urban Soils

**DOI:** 10.3390/ijerph16173191

**Published:** 2019-09-01

**Authors:** Hannah Heyman, Nina Bassuk, Jean Bonhotal, Todd Walter

**Affiliations:** 1School of Integrative Plant Science, Section of Horticulture, Cornell University, 134 Plant Sciences Building, Ithaca, NY 14853, USA; 2School of Integrative Plant Science, Section of Crop and Soil Sciences, Cornell University, 813 Bradfield Hall, Ithaca, NY 14853, USA; 3Department of Biological and Environmental Engineering, Cornell University, 232 Riley Robb Hall, Ithaca, NY 14853, USA

**Keywords:** organic matter, soil amendment, soil remediation, nutrient leaching

## Abstract

Poor soil health is a critical problem in many urban landscapes. Degraded soil restricts plant growth and microorganism activity, limiting the ability of urban landscapes to perform much needed ecosystem services. Incorporation of approximately 33% compost by volume into degraded soil has been proven to improve soil health and structure over time while avoiding the financial and environmental costs of importing soil mixes from elsewhere. However, additions of high volumes of compost could potentially increase the risk of nutrient loss through leaching and runoff. The objective of our study was to consider the effects of different compost amendments on soil health, plant health and susceptibility to nutrient leaching in order to identify ranges of acceptable compost characteristics that could be used for soil remediation in the urban landscape. A bioassay was conducted with Phaseolus vulgaris (Bush Bean) to measure the effect of nine composts from different feedstocks on various plant health parameters. Leachate was collected prior to planting to measure nutrient loss from each treatment. All compost amendments were found to improve soil health. Nutrient-rich, manure-based composts produced the greatest plant growth, but also leached high concentrations of nitrate and phosphorus. Some treatments provided sufficient nutrients for plant growth without excess nutrient loss. When incorporating as much as 33% compost by volume into a landscape bed, the optimal compost will generally have a C:N ratio of 10–20, P-content <1.0% and a soluble salt content between 1.0 and 3.5 mmhos/cm. These recommendations should ensure optimal plant and soil health and minimize nutrient leaching.

## 1. Introduction

Healthy soils have the potential to provide critical ecosystems services through processes including nutrient cycling, water infiltration, pollutant containment and carbon sequestration in addition to providing habitat for plants, animals and microorganisms. An important indicator of soil health is good soil structure. Healthy soil forms aggregates, creating pore space that can be filled by air and water and ease the growth of plant root systems. This process is made possible by the organic matter in the soil and the organisms that consume and transform it, providing the binding agents that help form soil aggregates [1,2].

Urban soil is generally characterized by the disturbance inflicted upon it by human activity such as burying of construction materials, soil importing, contamination and compaction, which can lead to imperviousness and soil sealing. Urban soil also tends to lack OM and, as a result, exhibits little to no microbial activity [3,4,5]. These characteristics make urban soil a poor habitat for plants and debilitate the growth of healthy urban ecosystems. To correct for this, the common practice in the landscaping industry is to remove and replace soils with specified soil mixes. These soil mixes are mined off-site and shipped to the desired location. This practice is costly and wasteful and does not address the underlying problem.

Compost amendment has been shown to improve physical, biological and chemical properties of many types of soil. It can decrease bulk density and increase porosity, OM content, microbial biomass, available water holding capacity, and structural stability [2,6]. However, compost is a highly variable product, which makes it difficult to assess quality and is, therefore, less appealing to landscape managers. Moving forward standardized testing protocols like the Test Methods for the Examination of Compost and Composting (TMECC), developed by the U.S. Composting Council [7] will be crucial in advancing the use of compost in the landscaping industry. In order to reap the full benefits of soil remediation with compost, one must fully understand the qualities of the compost being used, the qualities and limitations of the site and the desired outcomes.

A twelve-year study was completed at Cornell University in 2015, to measure the impacts of a soil remediation strategy on various soil quality indicators [8]. This strategy (The Scoop & Dump Method) consisted of physically fracturing compacted soils and incorporating large amounts of compost (33% by volume) to a depth of approximately 45 cm with the use of a backhoe or excavator. After planting bark mulch was added to the soil surface. The study found that, over time, remediated soils exhibited improved bulk density, increased active C and increased mineralizable N, as well as improved aggregate stability and available water holding capacity. Chen et al. (2014) and Rivenshield & Bassuk (2007) discussed similar effects of compost on soil health, however, only one type of compost was tested in each of these studies [1,9].

This study seeks to gauge the effects of different composts from different feedstocks on soil and plant health. Previous papers have assessed aspects of compost quality for horticultural purposes, often comparing one or two composts of the same feedstock to peat moss and soil alone. This paper’s contributions to the literature are two-fold. The first is the comparison of nine composts of different feedstocks with a wide range of characteristics. The second, is the inclusion of nutrient loss susceptibility, which is not often factored into compost quality [10].

A bioassay was conducted with *Phaseolus vulgaris* (Bush Bean) to measure the effect of composts from different feedstocks (animal manure, green waste, food scraps) at different concentrations (33% and 50% by volume) on various plant health characteristics (dry shoot weight, leaf area and leaf greenness). We also collected leachate from each treatment during the experiment to measure nutrient (N and P) loss from our different compost-amended soils. 

Nutrient leaching is a concern when high levels of compost are applied to landscapes before plant establishment or any time plants are unable to utilize large amount of N and P [2,11]. Organic amendments are often applied at N-based rates, which can lead to applications of P in excess of plant needs and increase the likelihood of nutrient loss in leachate or runoff [12]. Our objective was to consider soil health, plant health and susceptibility to nutrient leaching in order to identify a range of acceptable compost characteristics that could be used for soil remediation in the urban landscape. 

## 2. Materials and Methods 

### 2.1. Compost Selection

In the autumn of 2017 seventeen composts were collected from around New York State (Table 1). The composts collected were made from a variety of common compost feedstocks (e.g., manure, green waste, food scraps) and from a diversity of compost producers (e.g., farms, institutions, municipalities, private companies etc.). Approximately 75 L of compost were collected from each location. Two different batches of compost were collected from four of our producers. These batches were either prepared differently or a single company collected feedstocks from different locations. Most compost producers used a turned-windrow method of compost production [13].

A sample of each of the seventeen composts were brought to the Cornell University Nutrient Analysis Lab to be tested for C:N ratio by finding total carbon using the Combustion with CO_2_ Detection and Total Kjeldahl Nitrogen [7]. The compost samples were also tested for soluble salt content by measuring electrical conductivity using the slurry method as well as for OM% using the Loss on Ignition Method (LOI). Additionally, composts were tested for heavy metals, Arsenic, Barium, Cadmium, Chromium, Copper, Nickel, Lead and Zinc. All composts came back with levels below recommended maximum concentrations for soils in the Northeast according to the Cornell Waste Management Institute [14]. Composts were submitted to the Animal Health Diagnostic Center at Cornell University to be tested for coliform bacteria. None of the composts selected for the experiment contained detectable levels of coliform bacteria. It is critical, when selecting a compost, particularly for use in urban spaces, to test for harmful biological contaminants and heavy metals. All compost testing protocols were conducted according to the Test Methods for the Examination of Composting and Compost (TMECC) protocol [7] (Table 3). Based on those results we narrowed our study down to nine composts that represented a wide range of measured characteristics to use in our bioassay. Those nine composts were BOO, CC, CU, DL, WCE, OR, FF, BL and CG (Table 2). 

### 2.2. Soil Amendment and Testing

Arkport sandy loam soil (56% sand, 37% silt, 6% clay) was collected from the Bluegrass Lane Turf and Landscape Research Center in Ithaca, NY and sifted that soil through a 2.0 cm sieve. This soil was mixed with each of the selected composts to make the media for the bioassay. The Arkport soil and the eighteen compost-soil mixes were all sent to the Cornell Soil Health Lab for testing prior to the bioassay and then twice more during the course of the experiment (Table 3; Table A1). The soil alone was used as the control. The samples were stored in refrigeration at 4 °C (40 °F) prior to processing. Samples were analyzed for physical, biological and chemical indicators including available water holding capacity, aggregate stability, OM%, Autoclave Citrate Extractable (ACE) soil proteins, root pathogen pressure, soil respiration, pH, active C and extractible phosphorous using the Comprehensive Assessment of Soil Health: The Cornell Framework [14].

### 2.3. The Bioassay

The nine selected composts were each combined with the sieved Arkport sandy loam soil to serve as the growing media for the bioassay. Six repetitions were made of the following treatments, 100% soil (control), 100% compost for each of the nine composts, 50% of each compost with 50% soil by volume and 33% of each compost with 77% soil by volume. 

The bioassay was conducted in the greenhouse with *Phaseolus vulgaris* ‘Provider’ (bush bean) as our indicator species. Prior to planting, all treatments underwent a simulated heavy rain event in order to collect leachate. After that initial leaching, two *Phaseolus vulgaris* seeds were planted in each pot. Once the beans began to show true leaves if both plants had successfully germinated, one was disposed of. Pots were arranged in the greenhouse using a completely randomized design with six replicates and kept at 70 °F and 16-h days with overhead High Pressure Sodium High Intensity Discharge (HID) lamps. After germination, beans were watered with 150 mL of clear water every other day for the remainder of the experiment, excluding a second simulated heavy rain event conducted towards the end of the bioassay, for the purpose of collecting leachate. 150 mL of water was enough to keep the plants well-watered as they grew without allowing for more than slight leaching from the bottom of the pots. 

Beans were harvested 39–42 days after they were planted. SPAD 502 Plus Chlorophyll Meter (Konica Minolta, Ramsey, NJ, USA) was used to measure the “greenness” of the leaves. Leaf area was measured by taking a sample leaf from the second round of mature leaf growth from each plant and running it through a LI-COR 3100 leaf area meter (LICOR, Inc., Lincoln, NE, USA). Shoots were separated from roots and placed in labeled paper bags and dried at 70 °C for approximately two weeks after which dry shoot weight was measured. 

### 2.4. Leachate Testing

By putting each pot in a 5-gallon bucket, slowly filling the bucket with water until the water sat just above the level of the media in the pot and allowing it to soak for five minutes. Once pots were fully saturated, they were left for 24 h to reach “container capacity” (or field capacity). Container capacity of each pot was measured with a ThetaProbe Soil Moisture Sensor (Delta-T Devices Ltd., Cambridge, UK) and recorded. 150 mL of clear water was poured through the media and 40 mL of the leachate that came through the pot into the tray was collected and stored in a freezer for future analysis. 

Leachate samples were thawed overnight prior to testing. Prior to phosphorus (P) testing, 20 mL of each sample was collected and filtered through 45 μm filters. After filtering, samples were fed through an OI Analytical Phosphorus Analyzer Model 3000 (Xylem, Rye Brook, NY, USA) using the Ascorbic Acid Method of phosphate analysis [22]. Nearly all the leachate samples that were collected exhibited some coloration most likely due to high levels of tannins in the OM. This posed a challenge when using a colorimetric method of nutrient analysis because the pigment in the samples could possibly interfere with the absorption of the color reagent being measured. The darkest of the samples for phosphate analysis were diluted to overcome that interference. The darkest samples also showed levels of phosphorous that were well above the range of the instrument’s rating curve so dilution was necessary to receive an accurate reading. All WCE (poultry manure) compost mixes were diluted at a ratio of 100:1 and both the 100% BOO (cow manure) and 100% CU (mixed horse manure and green waste) at a ratio of 10:1 with deionized water. 

A colorimetric method developed by Hood-Nowotny et al. (2010) [15] was used to measure Nitrate and Ammonium in the leachate. This protocol was conducted using a Synergy^TM^ HT Multi-Mode Microplate Reader (BioTek^®^ Instruments Inc., Winooski, VT, USA). Ammonium was quantified by a colorimetric method based on the Berthelot reaction [16]. Nitrate was estimated after persulfate oxidation by reduction of nitrate to nitrite by Vanadium (III) chloride and a colorimetric determination of nitrite by an acidic Griess reaction [17]. Dilutions were necessary once again and the dilutions differed between the first and second leach events. Dilutions also differed for Nitrate and Ammonium tests to ensure that the reading fell within the range that could be accurately read by the micro-plate reader. Occasionally two different dilutions were made for a single treatment and an average was taken of the two readings.

### 2.5. Statistical Analysis

Statistical analyses were conducted using JMP pro 14.0 (SAS Institute Inc., Cary, NC, USA). Tukey HSD was used to compare mean values of the six repetitions in the bioassay and leachate collections. Linear regression analyses were conducted to determine correlation between compost and amended soil characteristics and plant growth as well as nutrient leaching. 

## 3. Results

### 3.1. Soil Quality

Compost amendment improved soil health regardless of feedstock type (Table 4) according to the Comprehensive Assessment of Soil Health completed at the Cornell Soil Health Lab [14]. Soil health tests were conducted on samples taken immediately after the incorporation of compost. Aggregate stability, OM percentage, soil respiration, ACE soil protein index and active C content increased for all amended soils compared to the control. The greatest increase in aggregate stability was observed in the 33% CU treatment. The greatest increase in OM% and Active C was observed in the 50% BL treatment. The greatest increase in the ACE soil protein index score was observed in the 50% CC compost treatment and the greatest increase in respiration was observed in the 50% CG treatment. Some of the amended soil mixes showed increased root pathogen pressure (50% CC, 50% and 33% CU). Manure-based compost mixtures generally exhibited higher values for root pathogen pressure, P and K content and soluble salt content and lower values for active C. For other soil characteristics like respiration, aggregate stability, available water holding capacity etc. feedstock type did not appear to have a noted effect. 

Surprisingly, twelve of the eighteen amended soil mixes exhibited either no improvement or slightly decreased available water holding capacities. AWHC of amended soils ranged from 16.2% to 36.7% compared to soil alone, which was 22.0%. This may be due to the larger particle size of the nine composts which reduced bulk density and available water, but more research would be required to verify this. All compost amendments increased the soluble salt content of the soil, from 0.03 of soil alone to 0.126 mmhos/cm, at the lowest (33% FF) to 2.924 mmhos/cm, at the highest (33% WCE). All but six of the amended soil mixes displayed extractable P concentrations higher than 25 mg/kg MMP (Modified Morgan Phosphorus), making them potential sources of nutrient loss [14,24]. The mixtures that did not were both concentrations (33% and 50%) of CG and OR compost as well as the 33% concentrations of the DL and FF composts. However, these mixes also showed the least impressive plant growth. The amended soil with the highest available P concentration was amended with 50% BOO containing as much as 180.137 mg/kg MMP, increased from the unamended soil concentration of 5.3 mg/kg of MMP.

BL leaf compost was the finest in texture with 63.2% of the compost particles smaller than 2.0 mm, followed by CC with 60.1% smaller than 2.0 mm. OR and DL composts were the coarsest in texture with 33.5% and 32.3% of compost particles being larger than 1.0 cm, respectively (Figure 1).

### 3.2. Relationship of Compost to Plant Quality 

The compost characteristics that had the greatest effect on plant growth were C:N ratio (Figure 2A), soluble salt content (Figure 2B), Phosphorus (P) content (Figure 2C) and Potassium (K) content. Soluble salt content of the amended and unamended soil had relatively strong positive correlations with both bean shoot weight and leaf area with R^2^ values of 0.57 and 0.51, respectively. Extractable P of the amended soil had a strong, positive correlation with plant growth. Leaf area and shoot weight had r^2^ values of 0.636 and 0.698, respectively, with increasing available P (Figure 2C). The positive correlation between K content of amended soils and plant growth was also strong with r^2^ values of 0.597 for leaf area and 0.652 for shoot weight. OM% of the composts and amended soils, alternatively, showed nearly no correlation with plant growth (Figure 2D). 

When composts with a C:N above 25:1 were incorporated into the soil, plant growth and chlorophyll concentration were reduced compared to the control. Shoot weight was reduced by as much as 80.9%, leaf area was reduced by as much as 77.3% and Leaf SPAD (greenness) was reduced by as much as 67.9% (33% OR compost) (Figure 3, Figure 4 and Figure 5). Beans grown using composts with a C:N close to 15:1 displayed the greatest shoot weight and leaf area (Figure 3 and Figure 4). C:N ratio and nitrate concentration of the compost had the greatest effect on chlorophyll concentration (Figure 5).

Manure-based composts outperformed the woody green waste-based composts, in terms of plant growth (Figure 3, Figure 4 and Figure 5). The only plant health parameter that did not differ based on compost type was root length. It is possibl that the size of the pot may have constrained root growth. BOO and CU had the greatest shoot weights (Figure 3), leaf areas (Figure 4) and shoot lengths. The treatments displaying the highest leaf SPAD (chlorophyll concentration) were the 33% CC, with a measurement of 35.6 and the 0% compost (control soil), at 35.5 (Figure 5). Bean plants grown in soil alone had highly concentrated chlorophyll because those plants were stunted in size with abnormally small leaves. CG and OR performed poorest in all categories. There were no plant growth measurements for the poultry manure compost (WCE) because the bean seeds were unable to germinate at any compost concentration. WCE compost had a soluble salt content of 17.585 mmhos/cm, a C:N ratio of 5.87, ammonium concentration of 3104.04 mg/kg and a P concentration of 63,260 mg/kg. We excluded the WCE compost from our analysis as an extreme outlier. Poultry manure compost is generally marketed for use as an agricultural fertilizer rather than as a soil amendment in landscape beds. 

### 3.3. Nutrient Leaching

There was a a relatively strong positive correlation between extractable P content of composts and amended soils and SRP content of leachate (Figure 6A). The same was not true for leached N, which displayed a weak correlation between nitrate found in the compost and nitrate content of the leachate (Figure 6B). Three of the four manure-based composts used (BOO, CU, WCE) leached significantly higher concentrations of SRP than the rest of the composts (Figure 7A). The composts that leached the greatest concentration of nitrate were CC and BOO followed by CU (Figure 7B). And the WCE compost was the only one to show significant amounts of ammonium leaching. We excluded WCE from our analyses as an outlier. The 100% CU compost treatment leached the highest concentration of SRP at 32.395 mg/kg SRP, while the 33% CU compost treatment leached only 3.985 mg/kg SRP. The 100% CC (food scraps) compost leached 340.417 mg/kg NO_3_, while the 33% CC compost treatment leached a far lower concentration, at 54.533 mg/kg NO_3_. Planting directly into 100% compost is not recommended. The leachate measured was collected prior to planting. It is possible that the high concentrations of nutrients found in the leachate would decrease significantly after even a short period of time, especially with the presence of actively growing plants to utilize some of the nutrients. 

### 3.4. Recommended Ranges

Using the data collected during the course of the experiment in concert with a review of the literature, a list of recommended ranges of certain compost characteristics was compiled (Table 5). These optimal ranges are intended to be utilized by compost producers, landscapers and practitioners in the industry when selecting a compost for on-site soil remediation. The test results of the composts used in the experiment were compared to the recommended ranges to evaluate how many would be considered suitable (Table 6).

## 4. Discussion

### 4.1. Testing Compost Quality

The composts selected for this experiment were chosen to encompass a range of compost characteristics and feedstocks. The objective was to choose composts that were commercially available and made from common feedstock sources to best reflect what landscape managers would have access to. C:N ratio, OM and soluble salt content were used as qualities to narrow down our composts from 17 to 9, anticipating that those characteristics would be the strongest indicators of compost quality. 

The initial compost quality parameters that must be met are those that indicate safety, such as contamination, maturity and traits like smell and presence of inert particles (trash). Those quality parameters apply to all compost regardless of their eventual use. Once those criteria are met, focus shifts to secondary quality parameters determined by the compost’s end use [31,32]. All nine composts used in the experiment tested well for maturity (above 6) when tested with a Solvita^®^ Basic Field CO_2_ test. However, the CG and WCE composts showed signs of immaturity. The CG (woodchips) and WCE (poultry manure) were included in the experiment to illustrate the extreme ends of the spectrum in terms of C:N and nutrient content. The woodchips were not, in fact, compost as they never underwent the composting process. The WCE poultry manure compost was composted but did not undergo a sufficient curing period. It was, instead, rapidly dried to prevent it from losing nutrients because it was intended to be marketed more as a fertilizer than as a compost. Its immaturity was one of the main reasons the bean plants failed to germinate in any of the WCE mixes. 

As for the secondary quality parameters, the intended end-use was urban or disturbed soil remediation with 33–50% compost by volume. These soil to compost ratios had been found to improve soil health and reduce bulk density over time [8,9]. At these volumes the main compost characteristics of concern were C:N ratio, soluble salt content and nutrient content (N-P-K). Nutrient content is not often included in compost specifications, although it is generally included in compost laboratory testing. We found that if nutrient leaching is a concern, nutrient recommendations are important considerations when specifying a compost. 

When testing compost, a recognized, consistent test protocol is critically important if one is to successfully adhere to written compost specifications and recommendations for use. We recommend compost producers and practitioners seek out labs that use TMECC, which was developed, with the assistance of many laboratories, by the U.S. Composting Council and modeled after the American Society for Testing and Materials (ASTM) [7]. Compost is an extremely variable product. Standardization of testing is a good way to mitigate uncertainty and increase universal understanding of a complex product that is often made from a mix of feedstocks and by a variety of processes. 

### 4.2. Soil Health

All compost amendments carried out in this experiment improved soil health according to the Comprehensive Assessment of Soil Health completed at the Cornell Soil Health Lab [14]. These improvements included increased OM, active C, ACE (Autoclave Citrate Extractable) soil proteins, respiration and nutrient content. These results either directly indicated an increase in microbial activity or suggested a potential for increased microbial activity. OM%, a measure of the biomass-derived carbonaceous material in the soil, is the main energy source for microorganisms. Active C is the portion of that food source that is the most easily accessible for microorganisms. Soil proteins represent the large pool of organically bound N in the soil OM that can be mineralized by microbes and made available for plant uptake [14]. An increase in OM from 2.2% in the control to as much as 8.85% with the addition of 50% BL compost was found including small positive correlations between increased OM% and respiration, aggregate stability and available water holding capacity (AWHC). Treatments with 50% compost tended to show higher values for those characteristics than those with 33% compost. 

Soil Respiration is a measure of carbon dioxide released from the soil due to microbial metabolic activity. The measurement of soil respiration integrates both abundance and activity of the microbial community. That activity includes nutrient cycling into and out of soil OM pools and N transformations like mineralization and nitrification. In this experiment, respiration increased with the addition of all compost types at all concentrations. Increased OM, active C, and soil proteins, increased microbial activity. The greatest respiration was observed in the CG compost treatments (50% and 33%) at 1.98 and 1.44 mg CO_2_, respectively. The CG compost did not display the highest OM%, protein content or active C content, however. The increased microbial activity might be due to the immature nature of the CG compost. There may have been more microbial activity because there was more potential for further decomposition. The 50% BL and 50% CC treatments displayed the highest OM% at 8.85% and 6.64%, respectively. They displayed the highest values for the ACE soil protein index at 20.26 (50% BL) and 23.40 (50% CC) as well as the highest active C contents at 1160.90 mg/kg (50% BL) and 951.82 mg/kg (50% CC). They correspondingly showed high levels of respiration at 1.15 mg CO_2_ in the 50% BL treatment and 1.31 mg CO_2_ in the 50% CC treatment. 

That increased microbial activity then influenced soil aggregate stability, water retention, nutrient cycling, and cation exchange capacity (CEC) [8,9,33,34,35,36,37,38]. Both 50% BL and 50% CC showed increased AWHC and increased aggregate stability. Compost can also inoculate soil that has been depleted of its microbial community. Pérez-Piqueres et al. (2006) found that incorporation of good quality composts may increase microbial biomass and enhance soil enzyme activity, although to what extent, depends on the compost and soil type [39]. It is likely some inoculation occurred in our experiment because respiration increased by a minimum of 81.25% and a maximum of 396.0% with the addition of compost (from 0.4 mg CO_2_ in the soil alone to 0.72 mg CO_2_ in the 33% OR and 1.98 mg CO_2_ in the 50% CG) shortly after incorporation. 

Aggregate stability increased by 19.4% to 97.4% with the addition of compost. Feedstock type did not seem to correlate with increased aggregate stability. Aggregate stability is greatly influenced by microbial activity as aggregates are held together by microbial products like polysaccharides, exudates and fungal hyphae. Treatments that displayed greater aggregate stability also showed greater plant growth such as BL, CU and BOO treatments (Figure 3 and Figure 4). CG treatments also displayed a high percentage of aggregate stability, but still displayed poor growth, most likely because large pieces of woody material were mistaken for aggregates during laboratory testing.

Available water holding capacity either stayed the same or decreased slightly in the majority of compost amended treatments. AWHC decreased by a maximum of 27% in the 33% WCE compost treatment. These results contradict most findings in the literature which cite increased AWHC with increased OM [1,6,8]. Saxton and Rawls (2006) found that soil OM between 0.5% and 8.0% has been proven to increase AWHC in silt loam soils [40]. However, despite OM% increasing for all eighteen of our treatments, only five displayed an increase in AWHC (33% and 50% BL, 50% CC, DL, BOO). The 50% BL (leaf compost) treatment displayed the greatest AWHC increase (68% up from the control), this treatment also showed the greatest OM%, 8.85%. The composts that displayed increases in AWHC (BOO, BL, CC) had larger percentages of fine particles (<2 mm). BOO, BL, and CC composts contained 57.6%, 63.2% and 60.1% particles that were <2 mm by dry weight, respectively (Figure 1). The treatments with the lowest AWHC were amended with OR, CU and CG composts which all displayed higher percentages of larger particles. OR, CU and CG composts contained 33.45%, 29.56% and 29.06% particles >1 cm by dry weight (Figure 1). With larger pores, water most likely drained away by gravity as it could not be held by adhesion as it is in finer soils. Soil quality measurements were taken immediately after incorporation. Over time, perhaps, once the compost could be broken down further by microorganisms, there may be a different results, however further research is necessary to confirm this. In subsequent soil tests taken four and seven months later, AWHC measurements fluctuated for all treatments (Appendix A). 

Amending urban soil with compost is a simple solution that could immensely improve the health of urban landscapes. Not only does compost improve the biological, chemical and physical health of the soil, it contributes to maintenance of that health long-term. Sax et al. (2017) found increases in active C and aggregate stability over the course of their 12-year study and continual decreases in bulk density over that same time period [8]. In urban areas, where landscapes get heavy use and often receive little regular fertilization, the long-term N availability that compost provides is particularly important [41]. Sæbø and Ferrini (2006) suggest an annual top-application of compost because it serves a dual purpose, providing nutrients and OM and assisting with weed suppression [34]. 

Considering only soil health, it appears nearly any compost would improve compacted soil with low OM, low microbial activity and high bulk density. But it is important to consider plant health and nutrient retention as well. 

### 4.3. Plant Health

Compost benefits plant growth indirectly, through remediating the soil and directly by providing nutrients immediately and continuously as it is transformed by microorganisms. However, because compost is a variable product, practitioners are often hesitant to utilize it as a nutrient source. Most compost specifications do not include nutrient recommendations, but in this experiment nutrient content was an important consideration, not only for determining plant growth, but also to gauge to what extent nutrients might be lost after application. C:N ratio, soluble salt content and P and K content were the compost characteristics that appeared to have the greatest effect on plant growth. 

The composts that performed the best in terms of plant health were BOO (cow manure-based compost), CU (horse manure and green waste compost), CC (food and green waste compost) and BL (leaf compost) (Figure 3, Figure 4 and Figure 5). These four composts had C:N ratios ranging from 11.5–17.2. Their soluble salt content ranged from 1.9–3.4 mmhos/cm. Their phosphorous content ranged from 0.73–2.20% and their K content ranged from 1.4–4.4% (Table 2). These results indicated that compost quality is not necessarily feedstock dependent. 

The C:N ratio range that proved optimal in this experiment was in line with what is often recommended in the literature for finished compost. According to Sikora and Schmidt (2001) the C:N ratio considered optimal for compost is based on the C:N ratio of stable soil OM which generally falls between 10 and 15 [25]. Chatterjee et al., 2013 stated in their review that the ideal ratio for a compost used as a growing medium was 12–18 [26].We found that a C:N ratio equal to or greater than 25 in the finished compost resulted in stunted growth and pale green color, most likely due to N immobilization which was confirmed by Brady and Weil (1999) [42]. Because we did not include a compost in our experiment with a C:N ratio between 17 and 25 we were unable to determine a maximum C:N ratio that would still allow enough available N for plant growth. Sikora and Szmidt (2001) and Sullivan et al. (2003) found that in composts with a C:N of 20 or less, 5 to 15% of total N became plant-available during the first year after application [25,43]. Because we chose beans as our bioassay species, we also must consider the effects of nodulation, which occurred in all treatments over the course of the bioassay. Despite nodulation, many plants exhibited yellow leaves and stunted growth suggesting that nodulation did not make up for low N in some of the treatments. 

Mupondi et al. (2006) and Warman and Termeer (1996) both utilized bioassays in the greenhouse to evaluate the use of compost mixes on plant germination and growth. Both found that a mix of nutrient-rich material composted with a carboniferous material resulted in the strongest plant growth. The compost that performed the best for Mupondi et al. was a pine bark and goat manure blend with a C:N ratio of 16, which is in line with our findings. Mupondi et al. found that composted pine bark alone immobilized N and resulted in stunted plant growth, much like our CG woodchips [44,45]. Warman and Termeer saw plant growth decline when greater than 50% compost was utilized in the growing media whereas many of our bioassay plants thrived in up to 100% manure-based compost [46]. Nutrient levels of the compost and nutrient requirements of the desired plants or crops will vary, but the literature seems to agree that a combination of nutrient-rich and carboniferous feedstocks provide for the best growing media. 

A low level of salinity is important in compost because it indicates the presence of nutrients in the form of cations and anions that are required for plant growth. High salinity, however, can inhibit germination and plant growth [47]. The treatments in this experiment with soluble salt content below 0.5 mmhos/cm resulted in poor growth and greenness, particularly when low salinity coincided with high C:N. There wasn’t sufficient data to offer a maximum safe soluble salt content based on this bioassay because a treatment with a soluble salt content between 3.4 mmhos/cm and 17.6 mmhos/cm was lacking. 17.6 mmhos/cm nhibited germination completely. The composts that performed the best in this study had soluble salt contents from 1.9–3.4 mmhos/cm. Much depends on plant selection and in urban landscapes the use of salt-tolerant plants is encouraged due to regular salting of roads and walkways in cities located in regions with cold winters. Much of the literature agrees that compost amendments that increase the soil soluble salt levels higher than 4 mmhos/cm can pose a risk to healthy plant growth [48], but many standard compost specifications set the maximum electrical conductivity levels as high as 10 mmhos/cm [49].

There was strong positive correlations between P and K content and plant growth this bioassay. This is not surprising because P and K are vital macronutrients. P is necessary for various plant processes such as photosynthesis, respiration, N fixation, root development, maturation, flowering, fruiting, and seed production [23]. The Modified Morgan method [50] of phosphorus extraction to measure available P was used in the growing media. This method tends to be less sensitive then other extraction methods such as Mehlich III, Bray-Kurtz P1 and Olsen [51]. However, extremely high levels of Modified Morgan phosphorus (MMP) were found in experimental treatments. The recommended 33% treatment of the composts that showed the best performance (BOO, CU, CC, BL) showed a range of MMP from 49.0–130.63 mg/kg MMP. Jokela et al. (1998) found the optimal range of MMP for field crops to be from 4.0 to 7.0 mg/kg. 4.0 mg/kg MMP was cited as the critical value and additions of P fertilizer were recommended for soil with MMP levels up to 7.0 mg/kg. In their paper, Jokela et al. characterized soil with MMP above 20 mg/kg as excessive [24]. All but three of our treatments (30% and 50% CG and 33% OR) exceeded 20 mg/kg MMP. Consequences of excess available P are far reaching, and P can remain in the soil far longer than N. For this reason, compost testing, site analysis and thoughtful timing of compost amendments are important considerations. Although the soil remediation method we are testing calls for 33% compost by volume, it may be wise to use 25%, if P leaching is a concern on the intended site. Amendments of 25% compost by volume have been shown to improve bulk density in compacted sandy loam soil [9]. 

Our results displayed both the positive and negative impacts compost amendment can have on plant growth. Type of compost and amount of amendment will depend on the needs of the plants, but compost is undoubtedly a sustainable, affordable nutrient source for plants in the landscape. 

### 4.4. Nutrient Leaching

Compost is less susceptible to nutrient losses during large rain events than inorganic fertilizers that are completely soluble, but the soluble nutrients in compost are still of concern [52]. Site and soil assessment are important steps to take prior to compost amendment, as are compost laboratory tests. 

In a drier area with deeper soil, composts made with a mixture of manure and some carboniferous bulking agent could be used safely. However, on a site with well-drained soil, particularly moist conditions, or a high risk of runoff, manure-based compost is most likely too high in P and will result in nutrient pollution. Hurley et al. (2017) suggest that ≤0.2% P be the definition of low P compost. Low P composts are primarily derived from yard or green waste, as opposed to composts derived from food scraps, manure, or biosolids [53]. The CG woodchips contained the lowest concentration of P of the composts we tested, with 0.22%. All non-manure-based composts used contained <0.9% P. Finding a compost with ≤0.2% P might be a challenge for compost users if leaching is a concern. 

Timing of compost incorporation is crucial, particularly when compost amendment is occurring before the landscape is installed. It would be unwise to leave the amended soil unplanted for long stretches of time because available nutrients will be lost without established plant uptake. Most compost specifications do not include N content, outside of the C:N ratio, and P content is generally omitted as well. When incorporating compost into soil at such large volumes it is necessary to include nutrient ranges in specifications to make informed management decisions. 

Borken et al. (2004) found composts rich in N can cause excessive nitrate leaching during the first one to two years after application. In their experiment, Borken et al. measured N leaching in a forested area and observed that the mineral soils acted as a significant sink for NO−3 and dissolved organic N [35]. This experiment confirmed that where there was deeper soil to catch nutrients as they leach, N and P-rich composts may be safer to use. 

Amlinger et al. (2003) discouraged the use of very large amounts of compost as a soil amendment, especially in well-drained soils. Nutrient leaching from compost-amended soils could exacerbate existing eutrophication problems, which threaten the health of coastal and freshwater ecosystems [52,54]. This danger is elevated when composts are applied in late autumn and winter when plants are not actively growing. Spring is the best time to apply compost, when plants can take up dissolved nutrients, so they don’t end up polluting groundwater [55]. 

There was a direct correlation between the concentration of MMP in the media and the concentration of soluble reactive phosphorus (SRP) found in the leachate (r^2^ = 0.79). According to Pote et al. (1996) the soil P extraction test that will best predict SRP loss depends on soil type. In their study using Captina silt loam, they found the distilled water and acidified ammonium oxalate (Sheldrick, 1984) extraction methods were the most accurate indicators of SRP in the leachate, although all the methods they used showed statistically significant correlations [56]. In 1999, Pote et al. came out with another study using three more ultisols to see if different methods would be more accurate with different soil types. They found several tests were good predictors (with an r^2^ > 0.90) for all three soils, including Mehlich III, Modified Morgan, Bray-Kurtz P1 and Distilled Water [51]. This confirms our results that MMP in the compost would be a good indicator of potential P leaching and a P extraction would be a valuable addition to regular compost laboratory analysis and specification. 

There was no compost measurement that correlated strongly with nitrate leaching on its own. A higher C:N ratio results in increased N immobilization and therefore reduces the threat of leaching. Increased C:N was negatively correlated with nitrate concentration in the leachate. However, the r^2^ was only 0.079. We assessed this relationship based solely on the 100% compost treatment, because we did not test for C:N in the soil mixes. Nitrate concentration in the compost was only slightly positively correlated with nitrate concentration in the leachate with an r^2^ of 0.145. We believe that a larger sample size could result in stronger correlations, however, more research is necessary to better predict likelihood of nitrate leaching from compost. 

## 5. Conclusions

Compost is a valuable renewable resource for rebuilding depleted soils, reducing compaction and reinvigorating disturbed landscapes. Our objective was to identify a range of acceptable compost characteristics that could be used for soil remediation in the urban landscape. Composts made from combinations of three main feedstocks, animal manure, green waste and food scraps were analyzed. Soil health, plant health and the potential of nutrient leaching were taken into account in the recommendations. Although all nine composts used in this experiment improved soil health, the green waste composts received the highest scores from the Cornell Soil Health Lab. Higher compost concentration (50%) tended to improve soil characteristics more than the lower concentration (33%).

Different results were found when plant growth was evaluated. The nutrient rich composts made from cow and horse manure and food scraps produced the largest, greenest plants. The woody composts were detrimental to growth, immobilizing all N that might otherwise be available to the plant. However, those nutrient rich composts that boosted plant growth, leached high levels of nitrate and SRP. 

By taking all the information collected from this research and experimentation into consideration recommended ranges for the ideal compost for urban soil remediation was developed. The main concerns were C:N, P% and soluble salt content. The ideal ranges were 10–20 for C:N ratio, 0.2–0.9% P and a soluble salt content between 1.0 and 3.5 mmhos/cm (Table 5). Composts that exhibit these characteristics tend to be combinations of several feedstocks, some richer in N and P like manure, food waste or grass clippings and others richer in carboniferous material. Moreover, these levels produced good plant growth with minimal nutrient leaching. There are a wide variety of composts available for growers and landscapers with distinct nutrient contents, nutrient leaching potential, bacterial community composition, and other qualities that vary by the feedstocks used and the process through which the compost was produced [26,57]. It is important to test compost qualities using a standard testing protocol such as the TMECC protocol.

When using compost as a soil amendment the safest approach is to understand site conditions, soil type and drainage, which will help improve plant growth and minimize nutrient leaching. As more is learned about compost properties including standardize testing and regulations, knowledgeable incorporation of compost will play a critical role in improving soil and plant growth in disturbed urban soils.

## Figures and Tables

**Figure 1 ijerph-16-03191-f001:**
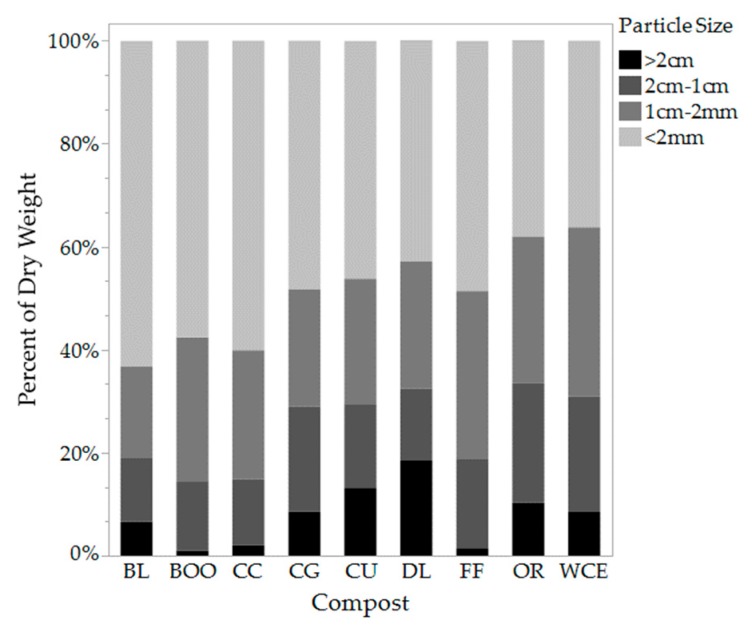
Composition of the compost particle size (>2 cm, 2 cm–1 cm, 1 cm–2 mm, <2 mm) by dry weight.

**Figure 2 ijerph-16-03191-f002:**
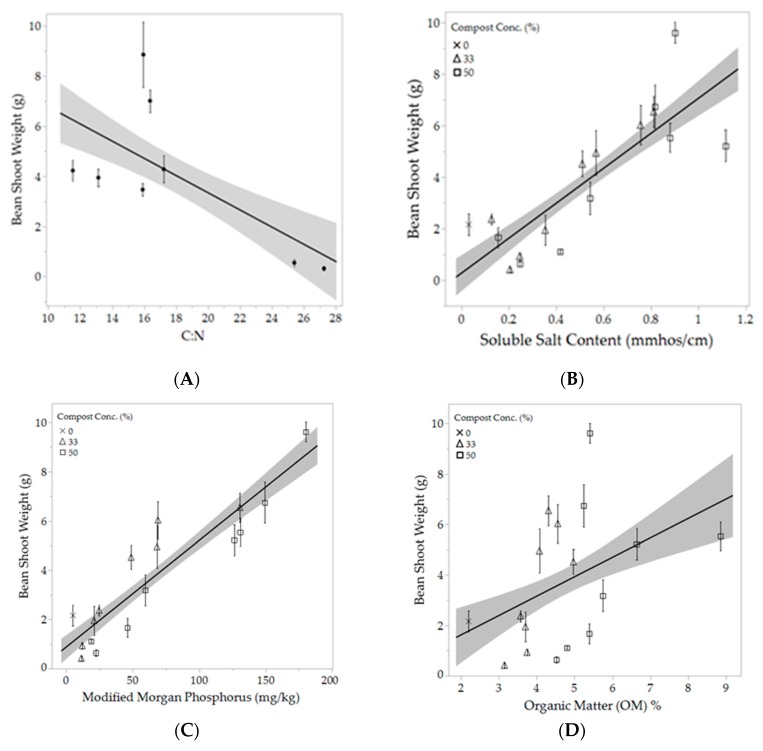
(**A**) shows the relationship between dry shoot weight of the bean plants and C:N ratio of eight of the nine compost types (excluding WCE). The black line represents the line of best fit; *R*^2^ = 0.359. (**B**) displays the relationship between dry shoot weight of the bean plants and soluble salts concentration of the amended and unamended soil for all compost types (excluding WCE); *R*^2^ = 0.570. (**C**) displays the relationship between extractable (available) phosphorus content of the amended and unamended treatments and bean dry shoot weight; *R*^2^ = 0.698. (**D**) displays the relationship between dry shoot weight of the bean plants and OM% of the amended and unamended treatments (excluding WCE); *R*^2^ = 0.157. The 100% compost treatments were excluded from Figure 2B–D for ease of interpretation. In all figures the shaded area denotes a 95% confidence interval. Each point represents the mean, error bars denote standard error (n = 6).

**Figure 3 ijerph-16-03191-f003:**
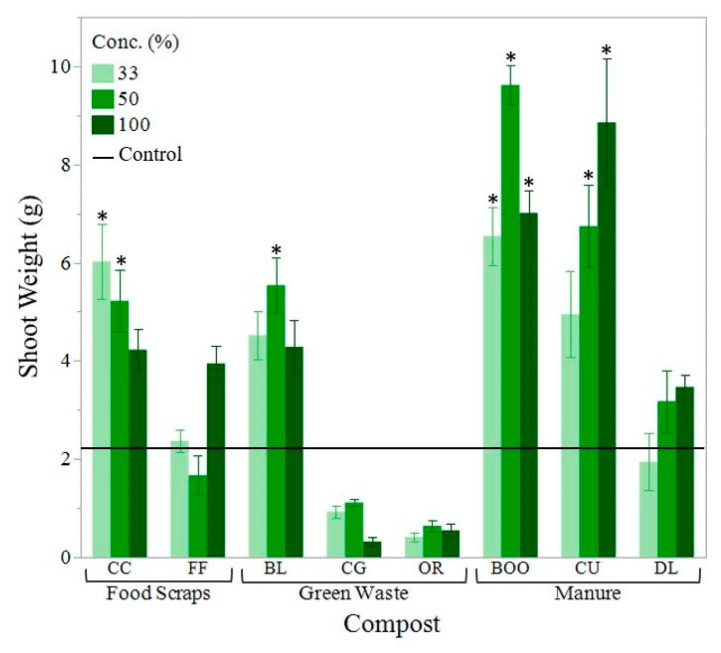
Mean bean plant dry shoot weight in grams by compost type. Compost concentration of the growing media shown from dark to light (100%, 50%, 33% compost). Horizontal solid black line indicates the mean shoot weight of the control (soil). Error bars denote standard error (n = 6). Stars (*) indicate significant difference from the control (α = 0.05).

**Figure 4 ijerph-16-03191-f004:**
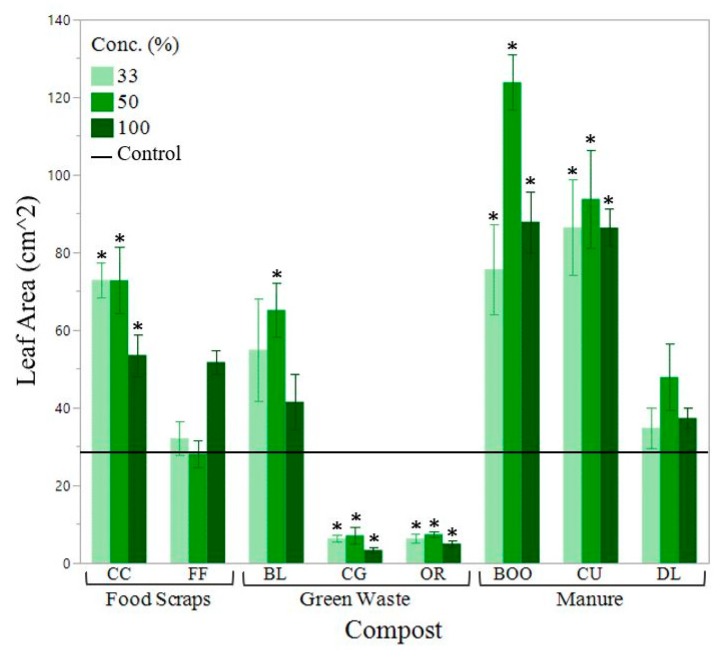
Mean bean plant leaf area in cm^2^ by compost type. Leaf area was taken for the second round of growth on bean plants. Compost concentration of the growing media shown from dark to light (100%, 50%, 33% compost). Horizontal solid black line indicates the mean shoot weight of the control (soil). Error bars denote standard error (n = 6). Stars (*) indicate significant from the control α = 0.05).

**Figure 5 ijerph-16-03191-f005:**
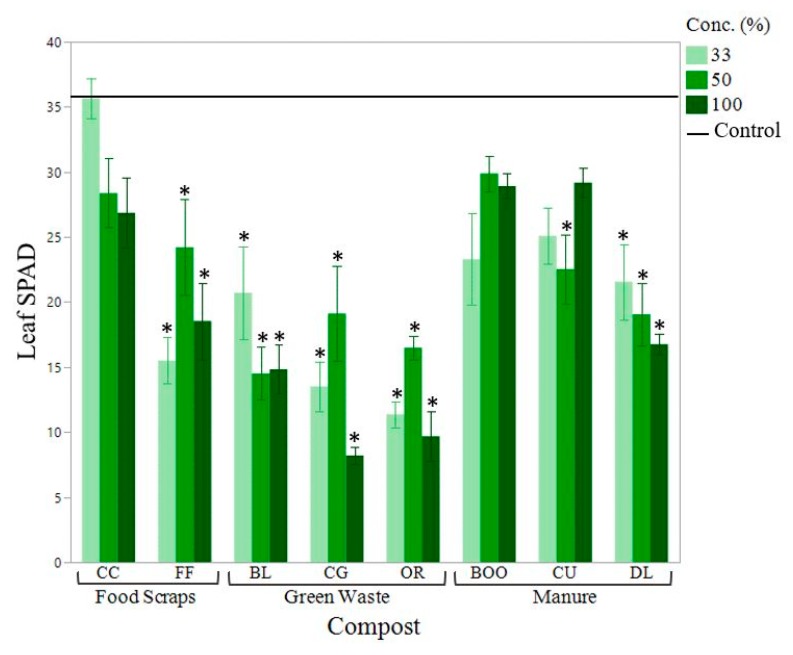
Mean bean leaf SPAD (greenness) by compost type. SPAD was taken using the second round of growth on bean plants. The mean of four separate measurements with the SPAD-meter was calculated for each plant. Compost concentration of the growing media shown from dark to light (100%, 50%, 33% compost). Horizontal solid black line indicates the mean shoot weight of the control (soil). Error bars denote standard error (n = 6). Stars (*) indicate significant from the control (α = 0.05).

**Figure 6 ijerph-16-03191-f006:**
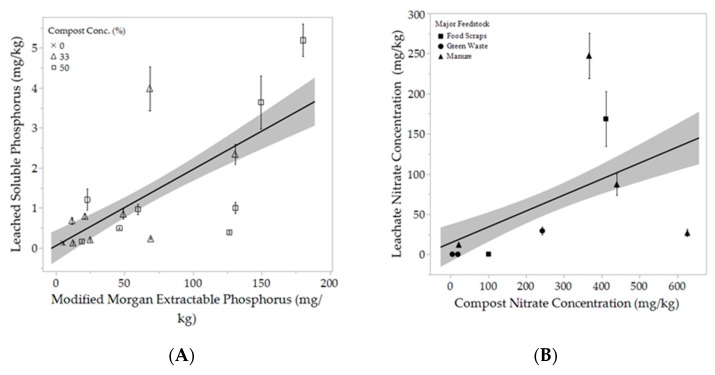
(**A**) displays the relationship between extractable (available) phosphorus content of the amended and unamended soil with concentration of soluble reactive phosphorus found in leachate. The black line represents the line of best fit; *R*^2^ = 0.417. The 100% compost treatment is excluded for ease of interpretation. (**B**) displays the relationship between nitrate concentration in the composts alone (100% compost). The manure-based compost in the lower left-hand corner is the WCE poultry manure compost which leached very little nitrate because the nitrogen in the compost was primarily in the form of ammonium. The black line represents the line of best fit; R^2^ = 0.145. In both graphs the shaded area denotes a 95% confidence interval. Each point represents the mean, error bars denote standard error (n = 6).

**Figure 7 ijerph-16-03191-f007:**
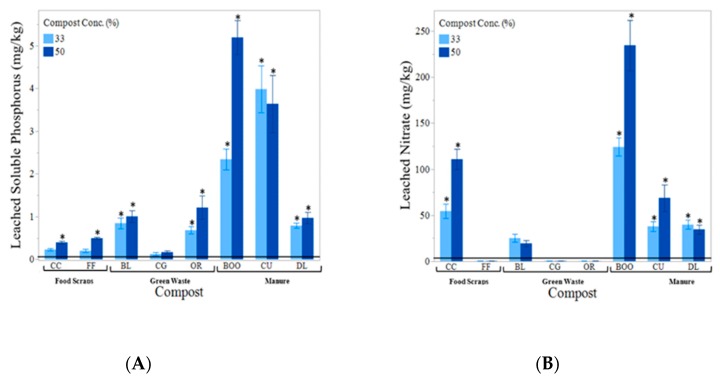
(**A**) shows mean soluble phosphorous found in leachate by compost type. Compost concentration of the growing media shown from light to dark (33% and 50%compost). The horizontal solid black line indicates the mean soluble phosphorus found in the leachate from the control (soil) equaling 0.122 mg/kg. (**B**) shows mean soluble Nitrate found in leachate by compost type. The horizontal solid black line indicates the mean nitrate found in the leachate from the control (soil) equaling 5.91 mg/kg. WCE compost and 100% compost concentration were excluded from both graphs. Error bars denote standard error (n = 6). Stars (*) indicate significant from the control (α = 0.05).

**Table 1 ijerph-16-03191-t001:** Seventeen composts collected from around New York State. Those in bold were selected for further experimentation.

Compost ID	Location	Major Feedstocks
**BOO**	**Greenwich, NY**	**Dairy Manure**
**VA**	Johnstown, NY	Dairy Manure
**CV**	Homer, NY	Dairy Manure
**CC**	**Trumansburg, NY**	**Food Scraps/Green Waste**
**CU**	**Ithaca, NY**	**Horse Manure/Green waste**
**DL**	**Stanfordville, NY**	**Horse Manure/Green Waste**
**WCE**	**Wolcott, NY**	**Poultry Manure**
**OCJ**	Jamesville, NY	Green Waste
**OCS**	Syracuse, NY	Green Waste
**OH**	Utica, NY	Green Waste
**ORC**	Orangeburg, NY	Green Waste (NY)
**OR**	**Orangeburg, NY**	**Green Waste (NJ)**
**FF**	**Staten Island, NY**	**Food Scraps**
**FY**	Staten Island, NY	Green Waste
**BS**	Bethlehem, NY	Green Waste (Screened)
**BL**	**Bethlehem, NY**	**Green Waste**
**CG**	**Ithaca, NY**	**Green Waste**

The bolded composts were the ones chosen for the experiment.

**Table 2 ijerph-16-03191-t002:** This data table shows the mean of two samples of each of the 9 composts tested at the Cornell Nutrient Analysis lab. Most of the tests shown above were taken according to the TMECC protocol [7]. nitrate and ammonium content were found using a KCl extraction.

Primary Feedstock	ID	Organic Matter (%)	Total Ash Content (%)	Total N (%)	Organic N (%)	NH4 (mg/kg)	NO3 (mg/kg)	P2O5 (%)	K2O (%)	Ca (%)	Mg (%)	Total Carbon (%)	C: N	Soluble Salts (mmhos/cm)	pH
Yard Waste	CG	35.27	78.33	0.82	0.82	10.94	5.15	0.22	0.57	5.20	0.75	22.32	27.26	1.01	8.25
	OR	72.91	78.76	2.83	2.83	2.06	19.85	0.53	1.53	4.33	0.70	71.63	25.40	1.32	7.52
	BL	65.60	81.58	2.93	2.91	2.86	242.88	0.73	1.79	4.88	0.94	55.62	17.22	2.64	7.54
Food Scraps	FF	25.50	95.39	1.62	1.61	2.98	100.90	0.63	0.84	4.16	1.10	20.82	13.13	0.85	8.06
	CC	24.23	95.70	1.64	1.60	5.98	410.80	0.82	1.41	2.22	0.57	17.40	11.53	1.94	7.66
Manure	DL	84.80	88.97	3.99	3.92	8.20	626.18	1.03	2.52	8.46	2.49	64.70	15.91	3.21	8.15
	CU	83.23	138.11	3.38	3.34	13.17	439.74	2.20	4.42	7.44	1.98	54.01	15.95	2.21	7.29
	BOO	52.59	82.34	2.50	2.46	4.08	366.34	1.07	2.25	4.54	0.78	42.03	16.37	3.41	7.67
	WCE	50.68	39.44	6.49	6.18	3104.04	22.18	6.33	3.33	11.1	0.84	35.61	5.49	17.59	6.73

**Table 3 ijerph-16-03191-t003:** Test methods used to test Compost, Amended and Unamended (control) soil and Leachate. Tests conducted on Compost were taken from TMECC protocol manual [7]. Tests conducted on soil were taken from the Comprehensive Assessment of Soil Health: The Cornell Framework [14].

Tests Conducted	Compost	Amended and Unamended Soil	Leachate
OM (%)	Loss on Ignition [7]	Loss on Ignition [14]	−
Total Ash Content	[7]	−	−
Total N (%)	Wet Oxidation Technique [7]	−	−
Organic N (%)	Semi-Micro Kjeldahl Technique [7]	−	−
Nitrogen-NH_4_	Colorimetric Method using a KCl extraction [7]	−	Colorimetric Method with a Berthelot Reaction [15,16]
Nitrogen-NO_3_	Colorimetric Method using a KCl extraction [7]	−	Colorimetric Method with a Griess Reaction [15,17]
P_2_O_5_	Inductively Coupled Plasma—Atomic Emission Spectroscopy [7]	Inductively Coupled Plasma—Atomic Emission Spectroscopy [14]	−
K_2_O	Inductively Coupled Plasma—Atomic Emission Spectroscopy [7]	Inductively Coupled Plasma—Atomic Emission Spectroscopy [14]	−
Ca	Inductively Coupled Plasma—Atomic Emission Spectroscopy [7]	Inductively Coupled Plasma—Atomic Emission Spectroscopy [14]	−
Mg	Inductively Coupled Plasma—Atomic Emission Spectroscopy [7]	Inductively Coupled Plasma—Atomic Emission Spectroscopy [14]	−
Total Carbon	Calcium carbonate equivilancy (Inorganic carbon) and Combustion with CO_2_ Detection (organic carbon)	−	−
C:N	Total Carbon:Total Nitrogen [7]	−	−
Soluble Salts	1:5 Slurry Method [7]	1:1 soil water suspension [14]	−
pH	pH electrode probe	pH electrode probe	−
AWHC	−	Difference of Field Capacity and permanent wilting point [14,18]	−
Aggregate Stability	−	Cornell Rainfall Simulator [14]	−
ACE Soil Protein Index	−	Colorimetric Method using a Thermo Pierce BCA protein assay [14,19]	−
Root Pathogen Pressure	−	Root Health Bio-assay [14]	−
Respiration	−	Incubation and CO2 capture [14,20]	−
Active Carbon	−	Colorimetric Method using Potassium Permanganate Oxidation [14,21]	−
Soluble Reactive Phosphorus	−	−	Ascorbic Acid Method [22]
Extractable Phosphorus	Modified Morgan Extraction [23]	Modified Morgan Extraction [23]	−

**Table 4 ijerph-16-03191-t004:** Unamended and amended soil characteristics. The soil used in all mixes is an Arkport sandy loam. Samples were taken immediately after mixing. Tests followed the Comprehensive assessment of soil health: the Cornell framework manual protocol by Moebius-Clune [14].

			Physical			Biological					Chemical			
Major Feedstock	ID	Compost Conc. (%)	Soil Texture	AWHC	Aggregate Stability (%)	OM (%)	ACE Soil Protein Index	Root Pathogen Pressure	Respiration (mg)	Active C (mg/kg)	P (mg/kg)	K (mg/kg)	pH	Soluble Salts (mmho/cm)
**Control Soil**	**S**	0	sandy loam	0.22	34.70	2.20	5.10	4.00	0.40	317.00	5.30	20.10	5.40	0.03
**Yard Waste**	**CG**	50	loam	0.20	63.58	4.80	15.35	3.00	1.98	753.37	18.85	300.12	6.90	0.42
	**CG**	33	sandy loam	0.20	60.12	3.74	11.67	3.00	1.44	562.00	12.32	178.98	6.74	0.25
	**OR**	50	sandy loam	0.22	48.68	4.52	18.26	3.33	0.90	732.10	22.59	338.02	6.51	0.25
	**OR**	33	sandy loam	0.19	50.40	3.15	12.78	3.33	0.72	553.14	11.60	182.86	6.19	0.20
	**BL**	50	loam	0.37	62.91	8.85	20.26	3.75	1.15	1160.90	130.82	955.36	7.09	0.88
	**BL**	33	sandy loam	0.24	57.14	4.97	15.81	3.00	0.83	918.15	49.00	429.28	6.75	0.51
**Food Scraps**	**FF**	50	sandy loam	0.21	49.89	5.38	11.83	4.00	1.29	827.78	46.09	315.95	7.26	0.15
	**FF**	33	sandy loam	0.20	50.13	3.57	13.15	3.75	0.99	629.33	24.89	199.10	6.92	0.13
	**CC**	50	loam	0.24	51.49	6.64	23.40	6.67	1.31	951.82	126.33	1192.96	6.67	1.12
	**CC**	33	loam	0.20	57.88	4.55	18.20	3.00	1.13	758.68	69.08	700.34	6.67	0.76
**Manure**	**DL**	50	loam	0.27	41.43	5.74	16.01	3.25	1.32	707.30	59.62	621.65	7.13	0.54
	**DL**	33	sandy loam	0.20	48.99	3.70	11.61	3.50	0.95	487.58	21.11	314.55	6.29	0.35
	**CU**	50	loam	0.19	64.05	5.24	14.39	5.00	1.08	570.86	149.41	1017.41	6.85	0.82
	**CU**	33	sandy loam	0.18	68.50	4.07	10.70	5.75	0.94	487.58	68.36	572.86	6.48	0.57
	**BOO**	50	sandy loam	0.24	58.54	5.40	15.92	3.00	0.98	664.77	180.14	1330.43	6.91	0.90
	**BOO**	33	sandy loam	0.19	60.38	4.31	11.79	4.00	0.85	579.72	130.63	965.05	6.25	0.81
	**WCE**	50	loam	0.17	64.12	6.91	53.25	5.80	4.90	918.15	1021.00	2515.26	6.69	2.04
	**WCE**	33	sandy loam	0.16	63.34	5.81	30.27	4.33	4.91	538.97	637.04	1780.03	7.05	2.92

**Table 5 ijerph-16-03191-t005:** Recommended ranges of compost characteristics for urban soil remediation. Recommended ranges were based on results found in the author’s experiment, confirmed and supplemented by the literature.

Parameters	Recommended Ranges	Units	References
pH	6.0–8.2	−	[10]
C:N	10–20	ratio	[25,26,27]
Organic Matter	>24	% dry matter	[10]
Soluble Salts	1.0–3.5	mmhos/cm	[10,28]
Total N	0.5–3.5	% dry matter	[29]
NO3-N	100–1000	mg/kg	[27]
NH4-N	<500	mg/kg	[27]
NH4:NO3	<10	-	[27]
P2O5	<1.0	% dry matter	[10,27,30]
K20	1.0–3.0	% dry matter	[10,30]
Particle Size	100% passing through 3 cm sieve 85% passing through 2 cm sieve 40–60% passing through 2 mm sieve	% dry matter	[10]

**Table 6 ijerph-16-03191-t006:** Results of tested composts. BL and CC composts fall within recommended ranges in every category and would therefore be recommended by the authors for urban soil remediation.

Primary Feedstock	ID	Organic Matter (%)	Total N (%)	NH4 (mg/kg)	NO3 (mg/kg)	P2O5 (%)	K2O (%)	C:N	Soluble Salts (mmhos/cm)	PH
YARD WASTE	CG	35.27	0.82	10.94	5.15	0.22	0.57	27.26	1.01	8.25
	OR	72.91	2.83	2.06	19.85	0.53	1.53	25.40	1.32	7.52
	BL	65.60	2.93	2.86	242.88	0.73	1.79	17.22	2.64	7.54
FOOD SCRAPS	FF	25.50	1.62	2.98	100.90	0.63	0.84	13.13	0.85	8.06
	CC	24.23	1.64	5.98	410.80	0.82	1.41	11.53	1.94	7.66
MANURE	DL	84.80	3.99	8.20	626.18	1.03	2.52	15.91	3.21	8.15
	CU	83.23	3.38	13.17	439.74	2.20	4.42	15.95	2.21	7.29
	BOO	52.59	2.50	4.08	366.34	1.07	2.25	16.37	3.41	7.67
	WCE	50.68	6.49	3104.04	22.18	6.33	3.33	5.49	17.59	6.73

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
