# Peer review of "Compost Quality Recommendations for Remediating Urban Soils"

_ijerph, 2019, doi:10.3390/ijerph16173191_

Round 1

Reviewer 1 Report

The article is on an important topic, but its presentation needs to be adjusted for the audience it is written for. In its current form, it may not be usable. It can be improved (please, see detailed comments within the manuscript):

1) create table(s), figure(s) to simplify the explanation of experimental design and materials and methods used instead of just describing them in the text.

2) create table(s), figure(s), flowchart(s) to simplify the explanation of results instead of just describing them in the text. How can urban practitioners utilize this study?

3) Avoid personalization in writing (e.g., we etc.).

Author Response

A table was created (with citations) and inserted on line 137 in the methods section summarizing the tests conducted on the compost, soil and soil mixes and on the leachate. We believe it was necessary to leave in the descriptions of the rest of the experimental design for the purpose of repeatability. An additional subsection along with two tables were inserted at line 495 in the results to summarize our results. One table details the ranges we recommend for certain compost characteristics. The second shows which composts in the experiment fall within those recommended ranges. All personalization was removed from the text according to the reviewer’s attached document with edits. Please see track changes.

Reviewer 2 Report

AN interesting study. But it fails to come across as especially novel - most of this was probably  done decades ago by botany or soils depts. I wondered about 2 things - heavy metals, which some composts may add (sewage being one); pathogens  - were the composts tested fow any harmful pathogens - urban soils come into contact with people..

I saw no control plots.....

Minor issues - title needs a : after Quality

Author Response

There are aspects of our experiment that have been previously studied. Many studies have used bioassays to assess differences in different compost types for horticultural use. However, our experiment’s novel contribution to the literature is twofold. First, we looked at a larger number of composts from different feedstocks. Second, we observed plant growth and nutrient leaching side by side. Considering nutrient leaching from finished compost in the assessment of its quality is a novel concept.  This is now described on line 74 in the introduction.

The composts used in our experiment were tested for heavy metals and biological contamination, but nothing of concern was detected. This explanation is now included in the paper starting on line 108 in the methods section.

We have six repetitions of a soil only control. The soil was tested, used in the bioassay and leachate was collected from this control. See line 131.

Regarding the title, we believe we are making compost quality recommendations so we prefer to leave out the colon.

Thank you for your insights.

Round 2

Reviewer 1 Report

Some revisions were made (e.g., tables summarizing results), but the article text needs to be improved. There is still a serious problem with numerous instances of "personalization" in writing (e.g., "We believe...", We wanted..." etc.). It is detrimental to scientific writing and should be corrected.

Author Response

All first person use of 'we' or "I' were removed from the text.
